# Characterizing Malicious Edges targeting on Graph Neural Networks

## Abstract

Deep neural networks on graph structured data have shown increasing success in various applications. However, due to recent studies about vulnerabilities of machine learning models, researchers are encouraged to explore the robustness of graph neural networks (GNNs). So far there are two works targeting to attack GNNs by adding/deleting edges to fool graph based classification tasks. Such attacks are challenging to be detected since the manipulation is very subtle compared with traditional graph attacks. In this paper we propose the first detection mechanism against these two proposed attacks. Given a perturbed graph, we propose novel graph generation method together with link prediction as preprocessing to detect potential malicious edges. We also propose novel features which can be leveraged to perform outlier detection when the number of added malicious edges are large. Different detection components are proposed and tested, and we also evaluate the performance of the final detection pipeline. Extensive experiments are conducted to show that the proposed detection mechanism can achieve AUC above 90% against the two attack strategies on both Cora and Citeseer datasets. We also provide in-depth analysis for different attack strategies and corresponding suitable detection methods. Our results shed light on several principles for detecting different types of attacks.

## 1 Introduction

Graph neural networks (GNNs) have been widely applied in many real-world tasks, such as drug screening (Duvenaud et al., 2015; Dai et al., 2016), protein structure prediction (Hamilton et al., 2017), and social network analysis (Hu et al., 2017). However, recent studies on adversarial examples, which are carefully crafted instances aiming to mislead machine learning models to make an arbitrarily incorrect prediction, have raised great concerns for the robustness of machine learning models. Given various applications of GNNs, researchers have proposed two types of attacks against GNNs by adding/deleting edges of a victim graph (Dai et al., 2018; Zügner et al., 2018).

Detecting such GNNs based attacks involves several challenges. First, such attacks focus on local graph properties and the manipulation of adding/deleting a small amount of local edges is not obvious enough to be detected by traditional Sybil detection methods (Bansal & Misra, 2016). Second, existing defense/detection methods against adversarial behaviors on machine learning models are not easy to be applied for detecting malicious attacks on GNNs. For instances, *adversarial training* is one of the most effective defense methods that generate adversarial instances and retrains the networks (Goodfellow et al., 2014b). However, it is impossible to label the malicious edges and retrain the GNNs model, since in the node classification tasks malicious edges contribute to feature vectors and influence several dimensions. Robust generative models are proposed to mitigate adversarial perturbation via denoising autoencoders and Generative Adversarial Networks (GAN) respectively(Goodfellow et al. (2014a); Meng & Chen (2017); Samangouei et al. (2018)). But subtle perturbation in graph structured data is hard to directly remove through generative models.

In this paper, we propose several detection methods against the GNNs based attacks, including Link prediction based detection (LP), Sub-graph link prediction based method (SL), Graph Generation based method (GGD), and Outlier detection based method (OD). Given a large graph which potentially contains a small number of malicious edges, we first test LP method as a baseline to evaluate how likely the adversarial edges can be inferred based on their prediction score. Then we propose to sub-sample several small graphs to perform detection. The underlying reason for this is that since

there are only a small amount of malicious edges, based on *law of large numbers*, most of the sampled sub-graphs will contain only benign edges. As a result, learning the edge prediction model based on these sub-graphs should be able to capture benign topology properties and therefore being able to tell apart malicious edges which they have never seen before. In addition, based on the "benign" topology properties, we propose to train a graph generation (GGD) model, which is able to predict benign edges and therefore compare the generated sub-graphs with existing ones to identify potential malicious edges. In certain cases, attackers are allowed to add more malicious edges as defined in (Zügner et al., 2018). Such slightly more obvious perturbation makes the sub-sampling step challenging for GGD. Therefore, we provide a list of intrinsic features that can be leveraged to perform outlier detection (OD). We empirically show that OD can achieve detection AUC above 90% when the victim node has a high degree. We also provide in-depth analysis for how the combination of our proposed detection primitive methods performs against existing two adversarial attacks on victim nodes with various properties. In summary, our contributions are listed below:

- We propose several detection methods to detect adversarial edges added by recent two attack strategies against GNNs. To the best of our knowledge, this is the first detection mechanism against the state-of-the-art GNNs based attacks.

- We propose a novel generation model for graph data and a filter-and-sample framework to train the generative model for the detection task.

- We explore attacks targeting on nodes with various properties and provide in-depth analysis for different attack strategies and corresponding detection methods.

- We provide several insightful features that can be leveraged to perform outlier detection against different attacks.

- Extensive experiments are conducted on both Cora and Citeseer datasets against the state-of-the-art attacks, demonstrating that the AUC is above 90% in most cases.

## 2 BACKGROUND AND PROBLEM STATEMENT

In this section, we will introduce the background on graph neural networks, our threat model, and our detection goal.

### 2.1 GRAPH NEURAL NETWORKS

Graph Neural Networks (GNNs) is a class of deep learning models on graph data $G = (V, E, X)$. A GNN calculates an embedding vector $\boldsymbol{\theta}_u$ of each node $u$ through an iterative process. At each iteration step it aggregates the information of itself and its neighbour to calculate a new embedding:

$$\boldsymbol{\theta}_u^{(k)} = f\big(\boldsymbol{x}_u, \boldsymbol{\theta}_u^{(k-1)}, \{\boldsymbol{x}_v, \boldsymbol{\theta}_v^{(k-1)}\}_{v \in \mathcal{N}(u)}\big)$$

Here $\mathcal{N}(u)$ denotes the neighbours of $u$ in the graph. The initial embedding $\boldsymbol{\theta}_u^{(0)}$ is the input node feature $\boldsymbol{x}_u$.

Many GNN models have been proposed and achieve good performance on various tasks, such as Graph Convolutional Networks (GCN) (Kipf & Welling (2017)) and Structure2Vec (Dai et al. (2016)). In this paper, we will focus on the GCN model. Let $\Theta^{(k)} = (\boldsymbol{\theta}_1^{(k)}, \boldsymbol{\theta}_2^{(k)}, \ldots, \boldsymbol{\theta}_{|V|}^{(k)})$ be the matrix of all node embedding vectors at step $k$. For GCN, the aggregation function is calculated as:

$$\Theta^{(k)} = \sigma\big(\hat{A}\Theta^{(k-1)}W^{(k)}\big)$$
$$\hat{A} = \tilde{D}^{-\frac{1}{2}}\tilde{A}\tilde{D}^{-\frac{1}{2}}$$

where $\tilde{A} = A + I_N$, $A$ is the adjacency matrix and $I_N$ is the identity matrix, $D_{ii} = \sum_j \tilde{A}_{ij}$, $\sigma$ is the non-linear activation function and $W^{(k)}$ is the trainable parameters in the $k$-th layer.

When applied to the node classification task, the model first calculates the embedding $\boldsymbol{\theta}_u$ for each node. Then the embedding vector of $u$ is fed into an output layer to calculate the probability vector

$$\boldsymbol{s}_u = softmax(W^{out}\boldsymbol{\theta}_u)$$

indicating the probability that node $u$ belongs to each class. During the training process, the goal is to minimize the cross-entropy loss for the prediction of nodes in $V_{train}$. During the evaluation process, the predicted class of each class is given by $y_u = \arg\max_y(\boldsymbol{s}_u)_y$.

**Semi-supervised node classification on graph** In a semi-supervised node classification task, we are given a graph $G = (V, E, X)$, where $V = \{v_1, v_2, \ldots\}$ denotes the set of nodes, $E = \{e_1, e_2, \ldots\}$, $e_i \in V \times V$ denotes the set of edges and $X = (x_1, x_2, \ldots, x_{|V|})$ represents the feature vector of each node. Each node $v_i$ has a label $c_i \in C$, but we only have access to a small subset of the true labels, i.e., $L_{train} : V_{train} \to C$ where $V_{train} = \{v_{i_0}, v_{i_1}, \ldots\} \subset V$ is the set of nodes which we know the true labels. We denote $V_{unk} = V \backslash V_{train}$ to be the set of nodes with unknown labels. Given $L_{train}$, we would like to infer the labels of $V_{unk}$. That is to say, we seek for a model to give a prediction $y_i \in C$ to each of the node $v_i$, and we would like to maximize the classification accuracy, i.e., $\sum_{v_i \in V_{unk}} \mathbf{1}\{y_i = c_i\}$.

## 2.2 ADVERSARIAL ATTACK ON GRAPH DATA

Recently, Dai et al. (2018) and Zügner et al. (2018) show that the GNN-based classification model can be fooled by malicious attackers. Suppose we have a node $v_{vic} \in V_{unk}$ which we call the *victim node* and our model can correctly predict the label $y_{vic} = c_{vic}$. An attacker's goal is to make small modification to the graph such that the prediction of the model is fooled. Formally, an attacker provides a graph $G'$ such that the prediction $y'_{vic} \neq c_{vic}$. The attacking method can be categorized into the following types: 1) **Feature attack**: $G' = (V, E, X')$. This means that an attacker made small modification to the feature vectors of some nodes. 2) **Direct Structure attack**: $G' = (V, E', X)$ where $\forall e \in (E \oplus E')$, $v_{vic} \in e$. Here $\oplus$ represents the symmetric difference of two sets. This means that the attacker can add or delete several edges in the graph and these edges are all directly connected to the victim node. 3) **Indirect structure attack**: $G' = (V, E', X)$ where $\forall e \in (E \oplus E')$, $v_{vic} \notin e$. This means that the attacker can add or delete edges in the graph, but none of the edges are directly connected to the victim node. 4) **Combination of the the above**.

## 2.3 GOAL OF MALICIOUS EDGE DETECTION

In this paper, we aim to develop a model to detect the malicious edges added by an attacker. That is to say, we assume that: a) the feature of the input graph is benign; b) in the structure attack $E \subset E'$. We make this assumption because: a) the detection of continuous malicious data has been investigated by many previous researchers, while detection of structural attack is a novel task; b) adding edge is usually a cheaper attack approach than deleting edge - for example, in an undirected citation network one can easily add edge by citing others but cannot easily delete edge if other papers have cited it.

Specifically, the model is provided with a malicious graph $G' = (V, E', X)$. We know that some of the edges are malicious, but we don't have any other clues about the attack information (e.g. the information of victim nodes and the model for attack). Given $G'$, the model should generate a score $s_j$ for each edge $e_j \in E'$ indicating how likely the edge may be a malicious edge. Thus, we can examine the suspicious edges manually to get rid of the malicious ones.

## 3 DETECTION APPROACHES

In this section, we will first introduce the proposed primitive methods for detection, followed by the proposed detection pipeline.

LinkPred (LP) The most straightforward idea is to perform a link prediction algorithm on the graph $G'$. A link prediction algorithm takes as input the node features and the adjacency matrix of a graph and output a probability for each node pair indicating how likely these two nodes should be connected. If the probability is low for the node pair of an edge, it is probable that this edge could be a maliciously added edge.

In practice, we first adopt a two-layer graph convolutional network to calculate the node embedding vector $\boldsymbol{\theta}_u$ for each node. Then for each pair node $(u, v)$ we use a bilinear hidden layer to calculate the probability

$$p_{u,v} = sigmoid(\boldsymbol{\theta}_u^\mathsf{T} W_{edge} \boldsymbol{\theta}_v)$$

The training process is to minimize the binary cross entropy loss of node pairs where the ground truth of node pairs with edge is 1 and pairs with no edge is 0. After the model is trained, we calculate the probability $p_{u,v}$ for all $(u, v) \in E'$. The smaller the probability, the more likely that the edge is a malicious one.

SubGraphLinkPred (SL) The idea of LinkPred is concise and clear. However, since malicious edges are contained in the graph $G'$, it is possible that the model may learn the wrong information

contained in those edges, thus leading to a wrong prediction. To alleviate this problem, we randomly sample small graphs from the original large graph and train a link prediction model on these sub-graphs. The intuition behind this method is that malicious edges are relatively rare in $G'$, and therefore based on the law of large numbers, most of the sub-graphs do not contain such malicious edges. Consequently, the effect of malicious edges on models would be mitigated since the probability of appearance of sub-graph with malicious edges is small. For each sub-graph $G'_i$, similar to LinkPred, we can calculate $\boldsymbol{\theta}_{u,i}$ for each node $u_i$. Then we yield the connectivity probability for each node pair $(u_i, v_i)$ as

$$p_{u,v} = sigmoid(\boldsymbol{\theta}_{u,i}^{\mathsf{T}} W_{edge} \boldsymbol{\theta}_{v,i})$$

The training procedure on the sub-graphs is same as LinkPred. For each node pair $(u_i, v_i)$ in the sub-graph, we can map it back to $G'$ as $(u_{back}, v_{back})$. In this way, after training, there would be multiple value on the probability of connectivity $(p_{u,v}^1, p_{u,v}^2..., p_{u,v}^i, ...)$ for each node pair $(u, v)$ because an edge may appear in various sub-graphs. In this case, we calculate the probability for all $(u, v) \in E'$ as $p_{u,v} = mean(p_{u,v}^1, p_{u,v}^2..., p_{u,v}^i, ...)$. Here $mean$ stands for the harmonic mean of all probability values. The smaller the probability, the more likely that the edge is a malicious one.

OutlierDetect (OD)   The intuition of this approach comes from an observation on the attacker behaviour during direct structure attack: in order to make the victim node be misclassified into another class (say class 1), the attacker tends to add many edges from the victim nodes with the nodes that are all in class 1. Hence, the class distribution of the victim node's neighbour can be quite diverged. In contrast, the classes in a benign node's neighbourhood should be quite uniform.

Inspired by this phenomenon, we propose an outlier detection model for the edges based on the class distribution of the neighbourhood nodes of an edge. In particular, we calculate the following features for each node: 1) Number of different classes in the neighbour; 2) Average appearance time of each class in the neighbour; 3) Appearance time of the most frequently appeared class in the neighbour; 4) Appearance time of the second most frequently appeared class in the neighbour; 5) Standard deviation of the appearance time of each class in the neighbour. For each edge, we calculate the above features for both nodes and concatenate them together, thus getting a 10-dimensional feature vector. We fit a one-class SVM with kernel over these edge feature vectors to detect the outliers. The trained model will calculate a score for each edge indicating the 'discrepancy' of it from the group. The larger the value, the more likely that the edge is a malicious one.

GraphGenDetect (GGD)   Similar to the SubGraphLinkPred approach, this approach based on the idea that if we randomly sample small sub-graphs from $G'$, most of the results will not contain malicious edges. As a result, we would like to propose a graph generation model which can generate the edges of a graph given the node feature $X$. We will train the generation model based on the randomly sampled small graphs. Then we use the trained model to predict which edges in $E'$ are least likely to be generated. These edges should be considered highly likely to be malicious.

We propose a graph generation model inspired by sequence generation approaches. At time step $t$, the model is given the node feature $X$ as well as previously generated edges, $E^{(t-1)} = \{e^{(1)}, e^{(2)}, \ldots, e^{(t-1)}\}$. The model will predict which edge is going to be generated by outputting a probability distribution over all the node pairs that haven't been connected yet:

$$e^{(t)} \sim P\big[(u,v)|(u,v) \notin E^{(t-1)}\big]$$

In practice, we use a GCN with bilinear output layer to calculate the probability. We first apply a two-layer GCN to calculate the embedding vector $\boldsymbol{\theta}_u$ for each node $u$. Then, for each node pair $(u, v)$, we apply a bilinear function $s_{uv} = \boldsymbol{\theta}_u^{\mathsf{T}} W \boldsymbol{\theta}_v$ to calculate the score. When deciding which edge to add at the next time step, the probability is calculated by the softmax of the scores of all the node pairs which have not been generated, i.e.:

$$P\big[(u,v)\big] = \text{softmax}\big(\big\{s_{uv}|(u,v) \notin E^{(t-1)}\big\}\big)$$

During the training process, at each time we will randomly generate a permutation $\pi$ and thus get an edge sequence $(e_{\pi(1)}, e_{\pi(2)}, \ldots, e_{\pi(|E_i|)})$ for each sub-graph $G_i = (V_i, E_i, X_i)$. Following the training pattern of classical sequence generation model, at each time step we fed in the current adjacency matrix $A^{(t)}$ and the node feature $X$ to calculate the score for each node pair $s_{uv}$. The training loss at each time step is a binary cross entropy loss. The ground truth label for node pair $(u, v)$ at time $t$ is 1 if $(u, v)$ is going to be added later in the sequence, i.e., $(u, v) \in (e_{\pi(t)}, e_{\pi(t+1)}, \ldots)$ and otherwise the label is 0.

In order to determine which edges are likely to be malicious, we first calculate the edge scores on sub-graphs. For each sub-graph $G_i$, we 1) we generate a permutation to get an edge sequence

$(e_{\pi(1)}, e_{\pi(2)}, \ldots, e_{\pi(|E_i|)})$; 2) at each time step $t$, feed in the adjacency matrix $A^{(t)}$ and node feature $X$; 3) at each time step $t$, calculate the scores $s_{uv}$ for the edges that have not appeared, i.e., $(e_{\pi(t)}, e_{\pi(t+1)}, \ldots)$; 4) the final score of an edge is the average of the scores which we calculate in all the time steps. Having calculated the scores of edges in each sub-graph, we can calculate the score of edges in the original graph by averaging them in the same way as we did in the Sub-GraphLinkPred approach. The smaller the score, the more likely that the edge is a malicious one.

**Filtering for GraphGenDetect**   As described above, GraphGenDetect is based on the idea that when we randomly sample small graphs, most will not contain the malicious edges on which we can rely to train a benign detection model. Another approach is that we can first filter away a proportion of edges in the original graph which seems to be suspicious and train our model on the filtered graph. If the malicious edges are indeed filtered away, all the sampled graphs should be benign during the training process and we can expect that our trained model is not affected by malicious edges. Note that the filtering algorithm can also be achieved by one of the above approaches because we can use it to calculate which edges are likely to be malicious one and filter them away. Therefore, we can adopt approaches that combine the above methods. For example, LinkPred + GraphGenDetect means that we first apply a link prediction algorithm to filter away the suspicious edges, and then train graph generation model to detect the malicious edges.

**Combined Pipeline for Detection - Ensemble (ENS)**   During the experiments we find that different approaches may perform differently on different victim nodes. In particular, we observe that LinkPred +GraphGenDetect performs the best on nodes with small degrees and OutlierDetect performs the best on nodes with large degrees. Therefore, our final pipeline is an ensemble model which averages the output of LinkPred +GraphGenDetect and OutlierDetect. Since the output of two approaches is different (LinkPred +GraphGenDetect outputs the probability that an edge is benign while OutlierDetect outputs the discrepancy of an edge from the majority), we need to normalize the two output: we first calculate the log-probability that an edge is malicious (for Graph-GenDetect it's 1 minus the probability of benign; for OutlierDetect it's the sigmoid of discrepancy) and then normalize the log-probability to [0,1]. Finally, we take the average of the two values to indicate how likely an edge is malicious.

## 4    EXPERIMENTAL EVALUATION

In this section, we will first introduce our utilized dataset and evaluation metrics, followed by the attack models, and performance analysis of the proposed detection methods.

### 4.1    EXPERIMENT SETUP

**Datasets and evaluation metrics**   We evaluate our detection model on both synthetic datasets and real-world citation networks. As for synthetic dataset, we generate two Erdos-Renyi graphs. and two scale-free graphs. The two Erdos-Renyi graphs are $G_{n,p_1}$ and $G_{n,p_2}$ with $n = 1000$ and $p_1 = \frac{\ln n}{n}$, $p_2 = \frac{2\ln n}{n}$. The two scale-free graphs are generated using Barabasi-Albert algorithms, with 1000 nodes and parameter $m_1 = 1$, $m_2 = 2$ respectively. We would like to assign node features and node labels that are related with the graph structures. Therefore, we first assign a 20-dimensional random feature $e_u$ to each node $u$. Then we let the node features to correlate with its neighbours by repeat:

$$e_u = \sum_{v \in \mathcal{N}(u)} e_v, \qquad e_u = e_u/||e_u||_2$$

where $\mathcal{N}(u)$ is the neighboring nodes of $u$. After repeat the process several times (in practice we repeat 3 times), we can get a hidden feature vector $e_u$ which is related with graph structures. The final node feature vector $x_u$ is a discrete random binary vector, the probability that the $i$-th bit of $x_u$ equals 1 is:

$$Pr[x_u^{(i)} = 1] = sigmoid(e_u^{(i)})$$

And the label of $u$ is $y_u = 1$ if $\sum_i x_u^{(i)} > 0$ and otherwise $y_u = 0$. The node classification accuracy can reach around 80% for ER graphs and around 75% for scale-free graphs.

The real-world citation network datasets we use are Cora (McCallum et al., 2000) and Citeseer (Giles et al., 1998). Cora has 2708 nodes and 5429 edges; Citeseer has 3327 nodes and 4732 edges. The node classification task on these two datasets is the same as in Dai et al. (2018).

For each dataset and attack approach, we select several victim nodes and perform the attacks to generate malicious edges. Then we perform the detection method on the new graphs and check whether the malicious edges can be detected. The evaluation metric is the area under ROC curve (AUC), which is a commonly used metric to verify the performance of a detection method.

**Attack models**    We evaluate attack models proposed by Dai et al. (2018) and Zügner et al. (2018). In particular, three types of attack are considered:

- Single-edge attack. This attack is proposed by Dai et al. (2018) where an attacker can add only one malicious edge to the graph.

- Multi-edges direct attack. This attack is proposed by Zügner et al. (2018) where an attacker adds several edges to the graph. The malicious edges are all connected to the victim node and the number of malicious edges should not exceed the degree of victim node.

- Multi-edges indirect attack. This is similar to the Multi-edges direct attack except that none of the malicious edges are directly connected to the victim node.

We follow several rules when selecting victim nodes. First, the attack must be successful on the victim node to fool the model. Next, we try our best to find successful attacks on victim nodes with different node degree to evaluate diverse victim nodes' properties. Finally, we choose victim nodes among those with the same degree uniformly at random to perform the detection. We observe that without considering detection, the Multi-edges direct attack is the most successful attacking model, followed by Single-edge attack and finally Multi-edges indirect attack. Therefore, we selected 20, 10, 6 victim nodes respectively for these three attack methods on real-world data. For synthetic data, we simply pick two victim nodes, one with the smallest degree and the other with the largest degree. The selected victim node degrees are shown in the appendix.

**Randomly adding edges**    We also try randomly adding edges as an alternative to adding edges maliciously. By this control experiments we want to see whether our approaches are detecting malicious behaviors or simply detecting anomaly in the graph. In particular, we add several (1/2/4/8/16) random edges to the graph at each time, and run our algorithms to see whether those random edges are detected or not.

**Detection methods**    We implement all the detection models in Pytorch (Paszke et al., 2017) except for the OutlierDetect where we use the sklearn toolkit (Pedregosa et al., 2011) for one-class svm. In order to sample subgraphs from the original graph, we iterate through each node and extract its two-hop neighbourhood as a subgraph. Thus, we can get a set of subgraphs whose cardinality equals the number of nodes in the original graph. For LinkPred, we train it for 20 epochs using Adam optimizer (Kingma & Ba, 2014) with learning rate 0.01 and at each epoch we sample among node pairs which are not connected so that the number of positive and negative labels at each epoch is the same. For SubGraphLinkPred, we train it for 200 epochs using Adam optimizer with learning rate 0.001 and we use the same negative sampling policy as LinkPred on subgraphs. For OutlierDetect we fit a one-class svm with radial basis function kernel. For GraphGenDetect we train it using Adam optimizer for 15 epochs with learning rate 0.001. We observe that the detection result of the GraphGenDetect model is reasonable fast (e.g. 5 epochs can be enough) while the result can differ a lot at each epoch. Therefore, during test time we evaluate the trained model from the 6th to 15th epoch and take the average scores as the final prediction. For combination of models (denote as 'model1+model2'), we filter 50% of the edges using the result of the first model and train on the second one.

## 4.2    Real-world Dataset Results and Dicussion

The results of our detection model on Cora and Citeseer dataset are shown in Figure 1 and 2 respectively. From the experiments, we obtain several interesting observations and we discuss them as follows.

**Detection performance varies on different attacks and victim nodes.**  As shown in Figure 1 and 2, LP and GraphGenDetect perform well on Single-edge attack, most of Multi-edge indirect attack and the first several cases in Multi-edge direct attack. The common property of these attacks is that there are not many malicious edges which are simultaneously connected to one node. The most effective approach for malicious edge detecting in such cases is the pipeline LinkPred +GraphGenDetect. It achieves an average AUC of over 0.91 for Single-edge attack.

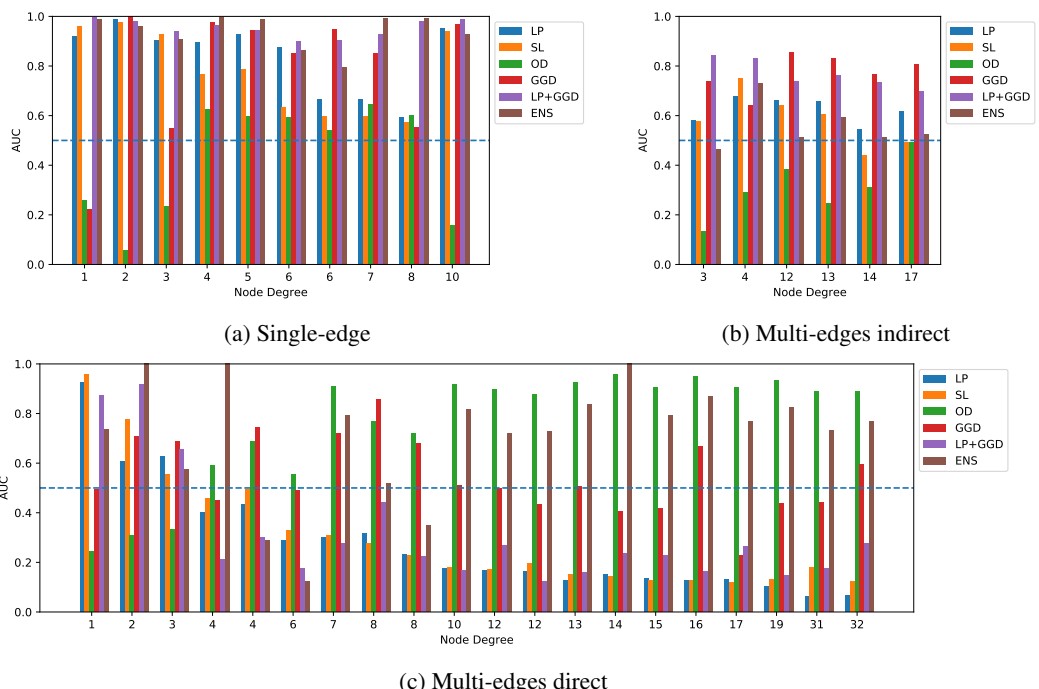

(a) Single-edge                    (b) Multi-edges indirect

(c) Multi-edges direct

Figure 1: Performance of detection on Cora dataset.

In the cases where many malicious edges are connected to a single node, which are most cases in Multi-edge direct attacks, approaches related to link prediction algorithms perform no better than random guessing. We attribute this to a principle of the collective power of malicious edges which one may express like this: when there are many malicious edges connecting to one node, they confirm the legitimacy of each other mutually. For example, suppose one node is originally connected to five nodes in class 1. If an attacker adds just one malicious edge that connects it to a node in class 2, this edge will seem abnormal and is easily detected. However, if the attacker instead adds five malicious edges in class 2, the legitimacy of each malicious edge will supported by the rest, and they will all be judged to be benign. By contrast, this collective power does not fool the OutlierDetect approach; its AUC is over 0.85 when five or more malicious edges are added, and the more edges the attacker adds the greater the likelihood of detection. For example, when ten or more edges are added the average AUC is over 0.9.

SubGraphLinkPred **doesn't outperform** LinkPred **obviously.** In SubGraphLinkPred, our intuition is that by sampling small graphs we can mitigate the effects of malicious edges. However, from the results, we see that these two approaches have roughly similar performance and trend. Therefore, we conclude that SubGraphLinkPred cannot outperform traditional LinkPred approach. Considering that the training and evaluation of SubGraphLinkPred will take more time, we use LinkPred as the method to combine with GraphGenDetect in later experiments.

**Combination of detection models can improve the performance.** When comparing the performance of the combination model LinkPred +GraphGenDetect with LinkPred in Single-edge attack, we see that as long as the LinkPred can filter out the malicious edge (i.e., AUC > 0.5), the combination model can significantly improve the detection performance. For Multi-edges attack, the combination can also improve the performance in most cases. This shows that our GraphGenDetect detection model can learn "benign properties" from normal graphs and improve detection performance. On the other hand, a direct GraphGenDetect can achieve fairly good performance in some cases where small number of malicious edges are added. This shows that GraphGenDetect is a good yet unstable model, so a filtering algorithm can be combined to reduce the variation and improve its robustness.

**Final pipeline is effective against malicious attack.** Our Ensemble model, which combines LP +GGD and OD, shows a good performance when either of the two approaches work well, i.e. when the node degree is small or quite large. For nodes with medium degree (e.g. 3, 4, 5), our approach cannot work well since both LP +GGD and OD cannot predict the malicious edges quite well.

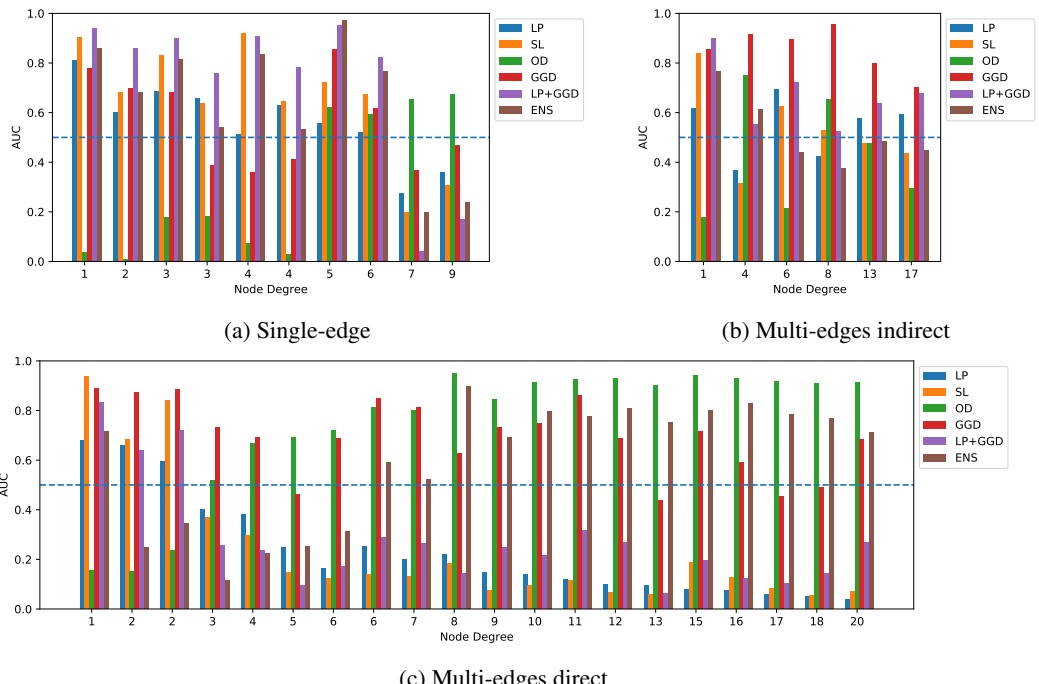

(a) Single-edge  (b) Multi-edges indirect

(c) Multi-edges direct

Figure 2: Performance of detection on Citeseer dataset.

**Detection performance differs in different dataset.** We observe that the performance of LP +GGD in Cora is slightly better than that in Citeseer. We owe it to the reason that the graph of Citeseer is more sparse than Cora (with more nodes and fewer edges). After filtering, the information contained in Citeseer dataset is less and therefore the performance decreases. One surprising observation is that direct GraphGenDetect approach achieves a fairly good result in Multi-edges attack in Citeseer. We presume that this phenomenon is also because of the sparsity: malicious edges tend to accumulate in a small neighbourhood of the victim node, and the sparsity induces that subgraphs of nodes far away from the victim node will not contain malicious edges. Therefore, the proportion of benign sub-graphs will increase and therefore the trained generative detection model can learn a better pattern of benign properties.

### 4.3 SYNTHETIC DATASET RESULTS

We also run the attack approaches and our detection framework on our synthetic ER graphs and scale-free graphs. The result is shown in Figure 3. Note that the scale-free graph by Barabasi-Arbert algorithms with parameter $m = 2$ is not in the figure. This is because the multi-edge attacks proposed in Zügner et al. will estimate the scaling parameters of the graph and requiring that the parameters should not change a lot after the attack. Therefore, the multi-edge approach cannot attack this graph successfully, so we do not need to defend against the attack.

We can see that the performance of our detection approach over synthetic graphs is as good as that over real-world data. That is: our LP +GGD approach can work well when there is only one edge; OD approach works well for multi-edge attacks when node degree is large; and when at least one of the approaches can work well, the Ensemble model can give a high detection performance. An exception is in the single-edge attack for BA1 graphs over the victim node with degree 6. Here the LP +GGD approach works badly. We investigate and find that the malicious edge connect two nodes with very high degree, so the LP is confused and fails to filter it away and therefore the GGD model is trained on the graph with malicious edge and learns badly.

### 4.4 RANDOM EDGE RESULTS

The experiment result of randomly adding edges to real-world graphs is shown in Figure 4 in Appendix. From the figures it is hard to tell a remarkable pattern of the performance over the edges, as no approach can perform very well to find the randomly added edges. Specifically, the detection AUC can hardly reach 80% for both Cora and Citeseer dataset, which is much worse than the detec-

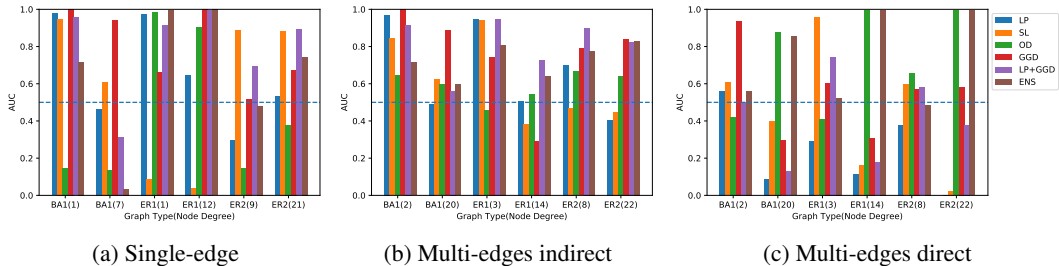

(a) Single-edge                     (b) Multi-edges indirect                     (c) Multi-edges direct

Figure 3: Performance of detection models on synthetic dataset. BA1 denotes the synthetic graph generated by Barabasi-Albert algorithms with parameter $m = 1$; ER1 and ER2 denotes the Erdos-Renyi graph with parameter $p_1 = \frac{\ln n}{n}$ and $p_2 = \frac{2 \ln n}{n}$ respectively. BA2 is not in the figure because the multi-edges attack fails on the graph.

tion of malicious edges. Therefore, we conclude that our detection approach captures the behaviour and can thus improve the robustness of node classification algorithms.

## 5 RELATED WORK

**Adversarial attack and defense methods.** Recently the robustness of machine learning models is widely studied, such as adversarial examples (Szegedy et al., 2013; Xiao et al., 2018a; Carlini & Wagner, 2017; Xiao et al., 2018b; Goodfellow et al., 2014b). As for discrete input spaces, such as text and graphs, the attack can be more difficult. So far there are two attacks proposed on graphs: Dai et al. proposed three attack methods on both node classification and graph classification problems and Zügner et al. developed an attack method targeted to a particular node based on greedy approximation scheme.

The researches with regard to defense methods on discrete input spaces are still limited. Though there are some researches on the topic of '*Anomaly Detection*' on graphs (Akoglu et al., 2015), the detection purpose is quite different: anomaly detection on graphs aims to find nodes/edges that differ a lot from others. While the two proposed subtle adversarial attack on graphs appear to contain too small magnitude of perturbation to be detected, and here we focus on graph neural networks instead of general graph analysis. In addition, Chen & Onnela (2018) studied on how to fit a good model over graph data when only one graph is available. They focus on the classification performance while our work focuses on the detection of malicious edges, and we have completely no supervision.

**Deep learning models for graph generation.** Generative models for graphs is important with rich applications including modeling social influences, analyzing chemical structures and building knowledge graphs. Recently the development of deep learning techniques fostered the progress towards generative models on graphs. Liu et al. brought up the issue of learning topological feature on graphs; Simonovsky & Komodakis and Grover et al. applied variational autoencoders (VAE) (Kingma & Welling, 2014) to generate graphs; Li et al. and You et al. modeled the construction of graph as a sequential process and used RNN for generation. You et al. proposed molecular graph generation through reinforcement learning. Instead of learning the topological features directly, Bojchevski et al. used GAN to generate random walks to capture the properties over graphs implicitly. Though there are abundant works on graph generation, they do not adopt the generative models to defending against attacks on graphs, which is the main focus of this paper.

**Robust collaborative filtering models.** Collaborative filtering, a common solution in recommendation tasks, can be viewed as a special method for graph data tasks. Several previous works have investigated into the robustness of collaborative filtering systems. Mehta & Nejdl (2009) detected malicious elements in social systems based on PCA and PLSA. Van Roy & Yan (2009) provided theoretical and empirical results on malicious manipulation over collaborative filtering algorithms and proposes a model which is robust against the manipulation. Nie et al. (2012) used joint Schatten $p$-norm and $l_p$-norm to better approximate the rank minimization problem and makes it robust to outliers. Recently, Hooi et al. (2017) proposed a optimization-based detection algorithms for fake connections which aims to confuse the collaborative filtering system. However, the relationship between user-item interactions in those tasks can merely be considered as a heterogeneous graph and we aim to find the malicious edges on the interaction graph with only one domain.

## 6 CONCLUSION

In this paper, we propose several approaches to detect malicious edges in graphs. We consider both state-of-the-art attack strategies against GNNs. We investigate into the attack and defense properties and find that different attack strategies led to different behaviour: some will add many correlated edges while some will add a single edge or un-correlated ones. For the first case, we propose a feature-based outlier detection model; for the second case, we propose a novel graph generation based model together with a filter-and-sample framework. In both cases, we show that the average detection AUC can reach above 90%. These results shed light on the design of robust deep learning models against attacks on graph data.

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

# A THE INFORMATION OF SELECTED VICTIM NODES

The attack method are different in different cases. In the Single-edge attack, one malicious edge is added; In Multi-edges attack, the number of added malicious edge can be up to the degree of victim node as defined in (Zügner et al., 2018).

| | Single-edge | Multi-edges direct | Multi-edges indirect |
|---|---|---|---|
| Cora | 1,2,3,4,5,6,6,7,8,10 | 1,2,3,4,4,6,7,8,8,10, 12,14,12,13,15,31,19,32,16,17 | 3,4,14,12,13,17 |
| Citeseer | 1,2,3,3,4,4,5,6,7,9 | 1,2,2,3,4,5,6,6,7,8, 9,10,11,12,13,15,16,17,18,20 | 1,4,6,8,13,17 |

Table 1: Degree of the selected victim nodes.

# B THE RESULT FOR ADDING RANDOM EDGES IN CORA AND CITESEER DATASETS

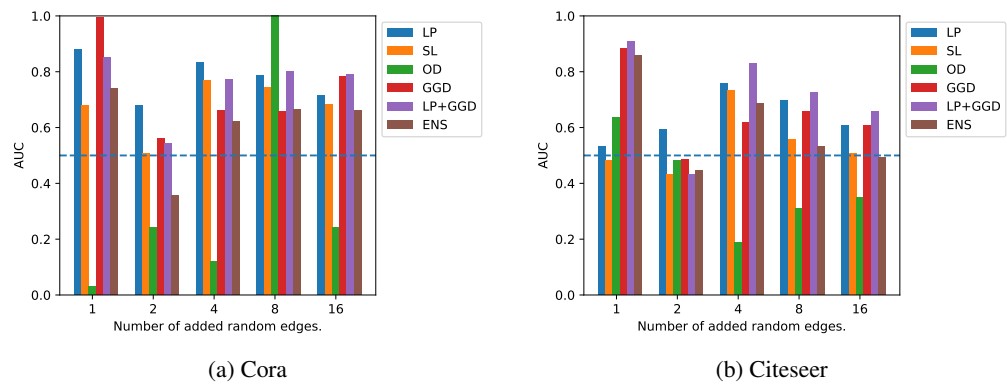

(a) Cora                      (b) Citeseer

Figure 4: Performance of detection models on randomly added edges on Cora and Citeseer.

