# OpenReview forum: "Characterizing Malicious Edges targeting on Graph Neural Networks"
_ICLR.cc/2019/Conference_

### Official Review · AnonReviewer1 · 2018-11-08
**First step on an important problem but hard to tell if its generally useful.**

**Rating:** 5
**Confidence:** 3

**Review:**

In this paper the authors present 4 methods for detecting outlier edges and nodes in graphs so as to prevent adversarial attacks on graph convolutional networks.  They demonstrate that their methods are accurate (through high AUC) in detecting edges added by two previous adversarial detection methods.

I think focusing on not just attacking GCNs but actually preventing them is awesome, and work of this sort should be highly lauded as I believe prevention is more difficult than attacks.  That said, it is hard to tell how general this work is.  The methods discussed follow fairly standard anomaly detection procedures (albeit with NN based models).  However, this leaves a few key open questions:

(1) To what degree do these methods address the particulars of GCN attacks?  This could possibly be addressed by better recapping the GCN attacks and explaining how these methods directly relate to those attacks.

(2) How robust are these methods?  While the intuition behind the methods at a high level seems reasonable, it is unclear if they provide any real robustness to an adversary.  Could the previous attacks be adapted if these detection mechanisms were known?  For example, I expect that adding edges that are high likelihood and maximally change the victim label would be an effective deception technique.  I believe a more thorough theoretical understanding of the robustness of the protection would make me more confident that these are broadly useful.  As of now, it seems very much data dependent.


Details:

GraphGen is worded weirdly -- you're not generating graphs, you're building a generative model for which you evaluate the probability that you would have generated an observed subgraph.

Robust MF has been studied and should be cited as well:

Benjamin Van Roy and Xiang Yan. Manipulation-resistant collaborative filtering systems. In Proceedings of the Third ACM Conference on Recommender Systems, RecSys ’09, pages 165–172, New York, NY, USA, 2009. ACM.

Bhaskar Mehta and Wolfgang Nejdl. Unsupervised strategies for shilling detection and robust collaborative filtering. User Model. User-Adapt. Interact., 19(1-2):65–97, 2009.

---

> ### Author Response · Authors · 2018-11-26
> **Response to Paper907 AnonReviewer1**
>
> We thank the reviewer for the valuable comments. Here are our responses to the concerns:
>
> Q1:[Unclear relationship with GCN attacks] To what degree do these methods address the particulars of GCN attacks?  This could possibly be addressed by better recapping the GCN attacks and explaining how these methods directly relate to those attacks.
> A1: Thank you for the interesting question. Basically, the motivation of this work is that: given GCN is currently widely applied, and recently two state-of-the-art attacks have been proposed against GCN models, therefore we aim to explore the possibilities of detecting such adversarial behaviors (added/deleted edges) in various scenarios. In addition, indeed there are attacks against other graph models, but there have been some heuristic detection methods to detect those such as sybil detection which has been studied for years. So here we aim to focus on these new types of attacks against GCN and it is a good idea for us to evaluate our detection method for other attacks in our future work.
> In addition, the GCN models exploit local patterns in a large graph to give reasonable predictions. In order to attack a GCN model, a malicious attacker can add edge to a node so that the local property of a particular node can be mixed with wrong information, thus leading to confusion. Therefore, our detection method actually leverages such property of GCN by sampling small subgraphs and train the detector which aims to avoid the adversarial impact of local malicious edges. We also exhibit a new experiment in the paper which adds random edges to the graph and use our methods to detect these edges. The result is not as good as the detection of malicious edges. This shows that our methods do capture the sophisticated adversarial behaviors targeting on graph models.
>
> Q2:[Lack of robustness analysis] How robust are these methods?  While the intuition behind the methods at a high level seems reasonable, it is unclear if they provide any real robustness to an adversary.  Could the previous attacks be adapted if these detection mechanisms were known?
> A2: It is hard for an adversary to do adaptive attack against our detection approaches since the detections here are based on graph structures which are non-differentiable. In addition, we have updated a new ensemble model which combines several of our proposed methods. This would make our pipeline harder to bypass since the attacker would have to bypass several different types of detection mechanisms. Nevertheless, we will study whether we can propose stronger adaptive attack without requiring gradient information, which is needed by current attacks, in the future work.
>
> Q3:[Unsuitble model names] GraphGen is worded weirdly -- you're not generating graphs, you're building a generative model for which you evaluate the probability that you would have generated an observed subgraph.
> A3: Thanks for the valuable suggestion. We have modified the name into GraphGenDetect since we are actually using generative model over graphs to detect malicious edges.
>
>
> Q4:[Insufficiency on related works] Robust MF has been studied and should be cited as well:
>   Benjamin Van Roy and Xiang Yan. Manipulation-resistant collaborative filtering systems. In Proceedings of the Third ACM Conference on Recommender Systems, RecSys ’09, pages 165–172, New York, NY, USA, 2009. ACM.
>
>   Bhaskar Mehta and Wolfgang Nejdl. Unsupervised strategies for shilling detection and robust collaborative filtering. User Model. User-Adapt. Interact., 19(1-2):65–97, 2009.
> A4: Thanks for the related work. Indeed MF-based collaborative filtering systems can be viewed as a special case of graph data, e.g. a bipartite graph between users and items. We have updated our related work by adding the suggested citations with discussions about studies on robust collaborative filtering systems.

---

### Official Review · AnonReviewer2 · 2018-11-08
**This manuscript tackles the interesting problem, but more improvement seems necessary in various aspects.**

**Rating:** 5
**Confidence:** 3

**Review:**

The authors address the interesting problem about the attack on the graph convolutional network model. The proposed method is developed under the assumptions about the attacking models sound simple but reasonable with proper references.

However, the proposed approaches mostly include the ideas about the detecting mechanisms instead of being formulated in some novel form. Given that the proposed algorithms do not leverage the underlying model structure very much, why the proposed algorithms are special to the graphical neural network is not very clear.

Also, the evaluations need to be further improved. It seems that victim nodes are carefully selected and fixed throughout all the experiments, but it limits the generalization about the performance of the proposed algorithms. Particularly, since the different detection algorithms perform differently on different datasets, more extensive evaluations are required along with the guideline of what detection algorithm we have to choose for the unsupervised setting.

*Details
- It will be great if the authors clearly describe what the proposed methods aim to defend. Basically, the values by protecting some victim nodes, regardless of what attacking models are assumed, will help the audience with better understanding. Some of content in Section 2.2 can be brought up in the introduction.
- In Section 3.1 and some other subsections, it seems to assume that the links in a given network are very clean but in reality there are a lot of noisy connections. How can we distinguish some random connections from malicious connections? Evaluation along with this question will be also useful.
- In Section 3.2, eventually, the ratio of malicious edges remains the same if the authors use random sampling. In that case, how SubGraphLinkPred helps is not very convincing.
- The detection algorithm seems to exist to detect malicious edges without supervision. In that case, how can we determine which method we should use given that detection performance differs in different dataset?
- It would be useful to compare with some existing malicious node/graph pattern mining algorithms such as Graph-Based Fraud Detection in the Face of Camouflage, Hooi et. al. even if the baseline method does not aim to directly solve the addressed problem. And also that literature needs to be cited.

---

> ### Author Response · Authors · 2018-11-26
> **Response to Paper907 AnonReviewer2**
>
> We thank the reviewer for the valuable comments. Here are our responses to the concerns:
>
> Q1:[The unclear relationship with GCN attacks] Given that the proposed algorithms do not leverage the underlying model structure very much, why the proposed algorithms are special to the graphical neural network is not very clear. It will be great if the authors clearly describe what the proposed methods aim to defend.
> A1: Thanks for the interesting question. The goal of our proposed methods is: given that graph neural networks are currently widely applied and recently two state-of-the-art attacks have been proposed against the models such as GCN, we aim to explore the possibilities of detecting such adversarial behaviors (added/deleted edges) in various scenarios. So here we aim to focus on these new types of attacks against graph neural network and it is a good idea for us to evaluate our detection method for other attacks in our future work.
> In addition, the graph models exploit local patterns in a large graph to give reasonable predictions. In order to attack a graph neural network, a malicious attacker can add edge to a node so that the local property of a particular node can be mixed with wrong information, thus leading to confusion. Therefore, our detection method actually leverages such property of GCN by sampling small subgraphs and train the detector which aims to avoid the adversarial impact of local malicious edges. We also exhibit a new experiment in the paper which adds random edges to the graph and use our methods to detect these edges. The result is not as good as the detection of malicious edges. This shows that our methods do capture the sophisticated adversarial behaviors targeting on graph models.
>
> Q2:[Limitation on the generalization of algorithms] It seems that victim nodes are carefully selected and fixed throughout all the experiments, but it limits the generalization about the performance of the proposed algorithms. More extensive evaluations are required along with the guideline of what detection algorithm we have to choose for the unsupervised setting.
> A2: We would like to emphasize that we did not cherry-pick the victim nodes. We are randomly picking the victim nodes according to its node degrees since we want to check how our approaches perform over nodes with various (large/small) degrees to evaluate the generalization of the proposed method. The ideal approach is to enumerate all the victim nodes and try our detection approach, but the computational cost would be too high. Therefore, we pick a subset of them randomly and evaluate our approach, and we think the subset size (node degree: 20/10/6) is large enough to show the performance of our models. In addition, we have proposed a uniform pipeline which combines our detection algorithms and we updated the performance of the uniform pipeline in the revision section 4.2.
>
> Q3:[Randomly adding edges] In Section 3.1 and some other subsections, it seems to assume that the links in a given network are very clean but in reality there are a lot of noisy connections. How can we distinguish some random connections from malicious connections? Evaluation along with this question will be also useful.
> A3: We thank the reviewer for the interesting question.
> First, in our experiments, we do not explicitly assume the given network is clean. For instance, the citation network dataset Cora and Citeseer we use is extracted from the real world, and there is no guarantee that the network is clean.
> In addition, based on the suggestion, we added additional experiments in section 4.4 which explicitly randomly add edges to the graphs. In figure 4 we show that our proposed detection method will only detect malicious connections rather than the random ones.
>
> Q4:[The effect of SubGraphLinkPred is unclear] In Section 3.2, eventually, the ratio of malicious edges remains the same if the authors use random sampling. In that case, how SubGraphLinkPred helps is not very convincing.
> A4: As long as the number of malicious edges is small, many subgraphs will contain no malicious edges at all. We hope that the link prediction model can learn better with these benign subgraphs, instead of simply learning on the large graph with malicious edges. In addition, we hope that the sampled small graphs will exhibit a similar pattern on which we can train a better graph neural network. In contrast, the single original graph is too large and therefore the pattern may be quite difficult to discover.

---

> > ### Author Response · Authors · 2018-11-26
> > **Response to Paper907 AnonReviewer2 (Part 2)**
> >
> >
> > Q5:[Lack of generalization] The detection algorithm seems to exist to detect malicious edges without supervision. In that case, how can we determine which method we should use given that detection performance differs in different dataset?
> > A5: In the updated version we have evaluated a uniform pipeline for detecting malicious edges, which combines several of our proposed models in section 3.6. We show that the uniform pipeline can detect adversarial edges with an average of over 80% AUC when victim node degree is very small or is large enough.
> >
> > Q6:[Insufficient comparisons with baselines] It would be useful to compare with some existing malicious node/graph pattern mining algorithms such as Graph-Based Fraud Detection in the Face of Camouflage, Hooi et. al. even if the baseline method does not aim to directly solve the addressed problem. And also that literature needs to be cited.
> > A6: Thanks for the related work. This paper aims to detect fraud links in collaborative filtering systems instead of graph models. We have updated our related work which discusses the robustness of collaborative filtering systems and have cited the paper in section 5.

---

### Official Review · AnonReviewer3 · 2018-11-09
**Important topic but significance can be improved**

**Rating:** 5
**Confidence:** 5

**Review:**

The study of detecting malicious edges in graphs is interesting and important.  However, the significance of the paper can be improved.  To properly test the detection performance, I recommend that the authors run experiments on various random graph models. Examples of random graph models include Erdos-Renyi, Stochastic Kronecker Graph, Configuration Model with power-law degree distribution, Barabasi-Albert, Watts-Strogatz, Hyperbolic Graphs, Block Two-level Erdos-Renyi, Multiplicative Attribute Graph Model, etc. That way we can learn on what types of networks the detection performance is better.  Also, in terms of detection models, I recommend that the authors try approaches that look for goodness of fit and model selection (e.g., see https://arxiv.org/pdf/1806.11220.pdf).

---

> ### Author Response · Authors · 2018-11-26
> **Response to Paper907 AnonReviewer3**
>
> We thank the reviewer for the valuable comments. Here are our responses to the concerns:
>
> Q1: [Significance of detection can be improved] To properly test the detection performance, I recommend that the authors run experiments on various random graph models. Examples of random graph models include Erdos-Renyi, Stochastic Kronecker Graph, Configuration Model with power-law degree distribution, Barabasi-Albert, Watts-Strogatz, Hyperbolic Graphs, Block Two-level Erdos-Renyi, Multiplicative Attribute Graph Model, etc. That way we can learn what types of networks the detection performance is better.
> A1: Thanks for the valuable suggestions. We have added additional experiments in section 4.3 for Erdos-Renyi graphs and Barabasi-Albert graphs which represent the scale-free graph family and apply our approach over these random graphs. The result shows that our approach is able to detect adversarial attacks within these random graphs as well.
>
> Q2:[Limitation on the generalization of algorithms] In terms of detection models, I recommend that the authors try approaches that look for the goodness of fit and model selection (e.g., see https://arxiv.org/pdf/1806.11220.pdf).
> A2: Thanks for the related work. We updated our related work and discuss the relationship with the suggested paper. In the paper, the authors also want to fit a good model given only one observed network. The authors focus on the classification tasks while we would like to detect malicious edges with no supervision.

---

### Author Response · Authors · 2018-11-26
**General Reply to the Reviewers.**

We thank the reviewers for their valuable comments and suggestions. Based on the reviews, we made the following update to our revision:
1. To demonstrate the generalization of the proposed method, we generate additional two graphs: Erdos-Renyi graphs and Barabasi-Albert graphs which represent the scale-free graph family and apply our detection approach over these random graphs in section 4.3. We show that our approach is able to detect adversarial attacks within these random graphs with high AUC scores.
2. We added the discussion for related work, talking about the robust collaborative learning systems in section 5.
3. We added additional experimental results on a unified pipeline of our proposed detection method in section 3.6 and 4.2, and we show that the unified pipeline can achieve usually achieve high AUC on different real-world and random synthetic graph datasets.
4. We added additional experiments on adding random edges and detect them in section 4.1 (randomly adding edges) and section 4.4. We show that our pipeline will only detect adversarial edges with high AUC instead of these random edges. This shows that our approach indeed captures the malicious behavior from sophisticated attackers rather than random noise (edges).

---

### Meta-Review · Area_Chair1 · 2018-12-14
**Clear reviewer consensus to reject**

**Confidence:** 4
**Recommendation:** Reject

**Metareview:**

All reviewers recommended rejecting this submission so I will as well. However, I do not believe it is fundamentally misguided or anything of that nature.

Unfortunately, reviewers did not participate as much in discussions with the authors as I believe they should. However, this paper concerns a relatively niche problem of modest interest to the ICLR community. I believe a stronger version of this work would be a more application-focused paper that delved into practical details about a specific case study where this work provides a clear benefit.